# SMAD4-Dependent Signaling Pathway Involves in the Pathogenesis of TGFBR2-Related CE-like Phenotype

**DOI:** 10.3390/cells13070626

**Published:** 2024-04-04

**Authors:** Yen-Chiao Wang, Olivia Betty Zolnik, Chia-Yang Liu

**Affiliations:** 1Edith Crawley Vision Research Center, Department of Ophthalmology, College of Medicine, University of Cincinnati, Cincinnati, OH 45267, USA; liucg@ucmail.uc.edu; 2Department of Anesthesia, School of Medicine, Washington University in St. Louis, St. Louis, MO 63110, USA; 3School of Optometry, Indiana University, Bloomington, IN 47405, USA; oliviazolnik@gmail.com

**Keywords:** TGFB, SMAD4, Corneal Ectasia, keratoconus

## Abstract

(1) Background: Our previous data indicated that disturbance of the Transforming Growth Factor beta (TGFB) signaling pathway via its Type-2 Receptor (TGFBR2) can cause a Corneal Ectasia (CE)-like phenotype. The purpose of this study is to elucidate whether the SMAD4-dependent signaling pathway is involved in the TGFBR2-related CE-like pathogenesis. (2) Methods: *Smad4* was designed to be conditionally knocked out from keratocytes. Novel triple transgenic mice, *Kera^rtTA^; Tet-O-Cre; Smad4^flox/flox^* (*Smad4^kera-cko^*), were administered with doxycycline (Dox). Optical Coherence Tomography (OCT) was performed to examine Central Corneal Thickness (CCT), Corneal Radius, Anterior Chamber and CE-like phenotype and compared to the littermate Control group (*Smad4^Ctrl^*). (3) Results: The OCT revealed normal cornea in the *Smad4^Ctrl^* and a CE-like phenotype in the *Smad4^kera-cko^* cornea, in which the overall CCT in *Smad4^kera-cko^* was thinner than that of *Smad4^Ctrl^* at P42 (*n* = 6, *p* < 0.0001) and showed no significant difference when compared to that in *Tgfbr2^kera-cko^*. Furthermore, the measurements of the Anterior Chamber and Corneal Radius indicated a substantial ectatic cornea in the *Smad4^kera-cko^* compared to *Smad4^Ctrl^*. The H&E staining of *Smad4^kera-cko^* mimics the finding in the *Tgfbr2^kera-cko^*. The positive immunostaining of cornea-specific marker K12 indicating the cell fate of cornea epithelium remained unchanged in *Smad4^kera-cko^* and the Proliferating Cell Nuclear Antigen (PCNA) immunostaining further indicated an enhanced proliferation in the *Smad4^kera-cko^*. Both immunostainings recapitulated the finding in *Tgfbr2^kera-cko^*. The Masson’s Trichrome staining revealed decreased collagen formation in the corneal stroma from both *Smad4^kera-cko^* and *Tgfbr2^kera-cko^*. The collagen type 1 (Col1a1) immunostaining further confirmed the reduction in collagen type 1 formation in *Smad4^kera-cko^*. (4) Conclusions: The aforementioned phenotypes in the *Smad4^kera-cko^* strain indicated that the SMAD4-dependent signaling pathway is involved in the pathogenesis of the CE-like phenotype observed in *Tgfbr2^kera-cko^*.

## 1. Introduction

Corneal stroma, serves as the major layer providing the most diopter for Visual Acuity (VA) in the cornea, is a highly organized collagenous matrix consisting of multiple collagenous lamellae and keratocytes [1,2,3,4,5]. The homeostasis of the corneal stroma requires proper formation of collagen, which is related to the SMAD4-dependent signaling pathway, a well-known pathway for Transforming Growth Factor beta (TGFB) signaling [6,7]. Disturbance in the corneal stroma can result in pathological changes, corneal abnormalities, and vision disorders. Among these disorders, Cornea Ectasia (CE) has the highest morbidity and is characterized by different patterns of corneal stroma thinning. Depending on the thinning pattern in the Optical Coherence Tomography (OCT), CE can be divided into keratoconus, keratoglobus and Pellucid Marginal Degeneration (PMD) [8,9].

Our previous study has established the Cause-to-Consequence rationale between Corneal Stroma TGFB Signaling and the CE-like phenotype, in which the deletion of the *Transforming growth factor receptor 2* (*Tgfbr2*) in the stroma cell, specifically the keratocyte in the animal model *Tgfbr2^kera-cko^* at postnatal day 1 (P1), can generate the overall cornea thinning, accompanied with corneal epithelium thickening and pathological thinning in the corneal stroma at puberty [10]. Furthermore, the classic clinical sign of Acute Corneal Hydrops (ACH) can be recapitulated in our animal model of *Tgfbr2^kera-cko^* in the eye-rubbing experiment. The similarities shared between *Tgfbr2^kera-cko^* and CE patients were manifested in diagnosis, tissue, cell, and microstructure levels, which indicated that the downstream signaling of the Transforming Growth Factor-beta Type 2 Receptor (TGFBR2) at the postnatal stage plays an important role in triggering the pathogenesis, and the corneal stroma can serve as the major layer contributing to the CE-like phenotype [10]. Our previous findings are consistence with genomic and RNA-seq studies in clinics, which implicated that TGFB signaling dysfunction might be associated with the etiology of CE [11,12,13,14,15].

However, although the Cause-to-Consequence rationale between TGFB signaling and the CE-like phenotype was established in our previous study, TGFBR2 serves as an initiation component for the general TGFB signaling, and the deficiency of this receptor can affect many downstream pathways including SMAD4-dependent and SMAD4-independent pathways [12,14,16,17]. Whether the SMAD4-dependent signaling pathway is involved in the etiopathogenesis still needs further investigation. Furthermore, in another report, the protein level of phosphorylated SMAD2/3 is elevated in the keratoconus stroma cell, which contradicts our established rationale [12,18]. Due to the complicated interactions within the general TGFB signaling, our previous study encountered difficulty in further elucidating the major signaling pathways behind the CE-like phenotype.

To elucidate whether SMAD4-dependent signaling is involved in the pathogenesis of the TGFBR2 deficiency-related CE-like phenotype in the aspect of Loss-of-Function, we generated a novel transgenic mouse strain, *Smad4^kera-cko^*, in which the *Smad4* is a conditional knockout from the keratocan (Kera)-positive keratocytes, the stroma cell, at postnatal day 1 (P1). The diagnosis approaches, basic science methods, and research strategies used in our previous study apply to this study to have a better comparison between these two conditional knockout mouse strains.

## 2. Method and Material

### 2.1. Mouse Strains

All the genetically modified mouse lines, *Kera^rtTA^, Tet-O-Cre (TC)* and *Smad4^flox/flox^,* have been previously described [19]. Compound transgenic mice were generated via natural mating of each mouse line. Most of the mice were bred at the Animal Facility of the School of Optometry, Indiana University, and few at University of Cincinnati. Experimental procedures for handling the mice were approved by the IACUC, Indiana University and University of Cincinnati. Animal care and use conform with the ARVO Statement for the Use of Animals in Ophthalmic and Vision Research.

### 2.2. Administration of Dox Chow

Mice were subjected to systemic induction by Dox chow (1 g/kg, Custom Animal Diets, Bangor, PA, USA). The Dox chow was given to *Smad4^kera-cko^* and *Tgfbr2^kera-cko^* at postnatal day 1 (P1) for postnatal development research and the measurements, including the OCT, OCE, HE stains, Fluorescent Immunostaining, and TEM, were performed at P42. For the embryonic research, the Dox chow was given on a breeder mating date and the data were collected at P1.

### 2.3. Hematoxylin and Eosin (H&E) Stain and Immunofluorescent Staining

Enucleated eyes and eyelids were fixed overnight in 4% PFA in PBS at 4 °C, followed by dehydration and paraffin embedding. De-paraffinized and rehydrated tissue sections (5 μm) were stained with Hematoxylin and Eosin and examined under a stereomicroscope (EVOSFL Auto, Life Technologies, Carlsbad, CA, USA). For immunofluorescent staining, tissue sections were subjected to antigen retrieval in sodium citrate buffer (10 mM sodium citrate, 0.05% Tween-20, pH 6.0) at boiling temperature. Cornea sections were then blocked with 3% bovine serum albumin (BSA) in PBS containing 0.05% NP-40 for 1 h at room temperature, then incubated overnight at 4 °C with the primary antibodies diluted in the same buffer. After three washes in PBST (PBS/0.1% Tween-20), slides were incubated at room temperature for 1 h with Alexa Fluor 488- or Alexa 555-conjugated secondary antibodies (Invitrogen, Waltham, MA, USA) and 1 μg/mL DAPI (Cat: #D3571; Molecular Probes, Inc. Eugene, OR, USA) as a nuclear counterstain, washed with PBST again, and mounted with Mowiol (Sanofi-Aventis U.S., Bridgewater, NJ, USA).

### 2.4. Corneal Thickness Examination

The central corneal thickness was measured by OCT using an iVue instrument (Optovue, Fremont, CA, USA). *Smad4^Kera-cko^* mice and littermate control were placed on a holder to keep the mouse corneas facing the OCT machine.

### 2.5. Genotyping

The identification of each transgenic allele was performed by Polymerase Chain Reaction (PCR) using tail genomic DNA as templates. PCR was performed by using the C1000TM Thermal Cycler (Bio-Rad Laboratories Inc., Hercules, CA, USA). After the initial step at 98 °C for 5 min, 40 cycles at 98 °C for 30 s, 65 °C for 30 s and 72 °C for 30 s were performed. Primer sets are listed below: *Kera^rtTA^* forward primer 5′ TGGTGGCTTGCTTCAAGCTTCTTC 3′, *Kera^rtTA^* reverse primer 1 5′ TATCCAACTCACAACGTGGCACTG 3′, *Kera^rtTA^* reverse primer 2 5′ GGAGTCTGCACTACCAGTACTCAT 3′, PCR product base pairs (bps); forward primer/reverse primer-1 knock-in 462 bps, forward primer/reverse primer-2 Wild-Type (WT) 389 bps. *Tet-O-Cre* forward primer 5′ GTCAGATCGCCTGGAGACGCC 3′, *Tet-O-Cre* reverse primer 5′ TCGCGAACATCTTCAGGTTCTGC 3′, PCR product 293 bps; *Smad4^flox^* forward primer 5′ TAAGAGCCACAGGGTCAAGC 3′, *Smad4^flox^* reverse primer 5′ TTCCAGGAAAAACAGGGCTA 3′, PCR product; knock-in 500 bps, WT 436 bps.

### 2.6. Statistical Analysis

A two-tailed Student’s *t*-test (Prism 10.2.1) was used to analyze the significance of the difference. *p*-value and the * number were automatically generated by Prism. The sign of ns represents “not significant” between two groups.

### 2.7. Transmission Electron Microscopy (TEM)

Tissue was fixed in 3% glutaraldehyde in 0.15 M sodium cacodylate buffer, postfixed in 1% osmium tetroxide in 0.15 M sodium cacodylate buffer, processed through a series of alcohols, infiltrated and embedded in LX-112 resin. After polymerization at 60° for three days, ultrathin sections (120 nm) were cut using a Leica EM UC7 ultramicrotome and counterstained in 2% aqueous uranyl acetate and Reynold’s lead citrate. Images were taken with a transmission electron microscope (Hitachi H-7650, Tokyo, Japan) equipped with a digital camera (BioSprint 16, Frankfurt, Germany).

### 2.8. Optical Coherence Elastography (OCE)

The corneas were collected from *Smad4^kera-cko^* and *Smad4^Ctrl^*. After the collection, the eyeballs were fixed in 4% PFA for 1 min to inhibit the enzyme reaction and then stored in the cornea transfer buffer Optisol-GS^®^ before OCE examination. The Wave Speed, Young’s Modulus, and Viscosity were measured at 10 mmHg IntraOcular Pressure (IOP).

## 3. Result

The SMAD4-dependent signaling pathway is one of the major pathways in TGFB signaling administering collagen formation in corneal stroma via controlling the activity of keratocytes [7]. Our previous study established the Cause-to-Consequence rationale between corneal stroma TGFBR2-deficiency and CE-like Phenotype [10]. Whether the CE-like phenotype is related solely to decreased collagen formation or more severe pathological changes disturbing the ECM organization remains elusive. To address this topic, whether SMAD4-dependent signaling can generate a similar CE-like phenotype as found in *Tgfbr2^kera-cko^* in the aspect of Loss-of-Function becomes critical.

Herein, we generated a novel transgenic mouse strain, *Smad4^kera-cko^*, in which the *Smad4* is deleted from the keratocyte at postnatal day 1 (P1) upon Dox administration (Figure 1 A). OCT, stereo microscopy, iCare Tonolab, H&E staining, Immunofluorescence, and TEM were applied to *Smad4^kera-cko^* and *Smad4^Ctrl^* at P42 to have a better phenotype comparison between *Smad4^kera-cko^*, *Tgfbr2^kera-cko^* and their control group mouse strains, respectively.

### 3.1. OCT Revealed a General Cornea Thinning in Smad4^kera-cko^ and Tgfbr2^kera-cko^

OCT revealed greater corneal thinning in *Smad4^kera-cko^* compared to those in the control group (Figure 1A–C). In detail, OCT revealed that instead of forming normal Central Corneal Thickness (CCT) of 113.00 ± 0.58 μm (Mean ± SEM) in the littermate controls at P42, the conditional knockout of *Smad4* in keratocytes resulted in a thinner cornea with a CCT of 77.00 ± 0.58 μm (*n* = 6, ****, *p* < 0.0001). (Figure 1B,C,F) Moreover, *Tgfbr2^kera-cko^* also exhibited a significantly thinner cornea with a CCT of 76.00 ± 1.32 μm compared to the control group with a CCT of 115.00 ± 0.68 μm (*n* = 6, ****, *p* < 0.0001). (Figure 1D,E,G).

We further compared the CCT of *Smad4^kera-cko^* and *Smad4^Ctrl^* to those of *Tgfbr2^kera-cko^* and *Tgfbr2^Ctrl^* to elucidate whether there is a significant difference between each with Dox induction during postnatal development. The OCT showed no significant difference between the conditional knockout groups of *Smad4^kera-cko^* and *Tgfbr2^kera-cko^*, (*n* = 6, ns) (Figure 1H) and control groups of *Smad4^Ctrl^* and *Tgfbr2^Ctrl^*. These data suggested the possibility that the dysfunction of SMAD4-dependent signaling is involved in the generation of TGFBR2-related CE-like cornea thinning.

### 3.2. Ectatic Cornea with Normal IOP Reading

To quantify the level of ectatic cornea, we used the perfect circle to measure the radius of the corneas via OCT images to check the expansion and protrusion situation in *Smad4^kera-cko^* and *Smad4^Ctrl^* (Figure 2A,B). OCT showed that, instead of showing a normal-sized cornea with a radius of 1060.00 ± 3.65 μm in the littermate controls at P42, deletion of *Smad4* in keratocytes resulted in ectatic cornea with a radius of 1151.56 ± 2.79 μm (*n* = 6, ****, *p* < 0.0001) (Figure 2A,B,D). Furthermore, we measured the distance between the lens and posterior cornea to see whether the Anterior Chamber was enlarged in the *Smad4^kera-cko^* compared to *Smad4^Ctrl^*. A significant Anterior Chamber enlargement was noticed in the *Smad4^kera-cko^* with a distance of 375.00 ± 9.13 μm compared to the control group with 341.67 ± 5.27 μm (*n* = 6, *, *p* = 0.013) (Figure 2A–C). We further used the iCare Tonolab to measure the IntraOcular Pressure (IOP) in *Smad4^kera-cko^* and the control group. The IOP reading from iCare Tonolab showed no difference between the two groups (Figure 2E).

### 3.3. Decreased VP Can Be Found in Ectatic Cornea

In clinics, decreased Visual Acuity (VA) is one of the major symptoms in CE patients. In this study, we desire to check whether the ectatic cornea can also result in decreased Visual Precision (VP) in *Smad4^kera-cko^*. A stereo microscope was applied to our animal models of *Smad4^kera-cko^* and *Smad4^Ctrl^* to have a better understanding of the ocular surface conditions.

The stereo microscope showed transparency in both groups with a well-developed ocular appendage (Figure 3A,B). Furthermore, the sharp reflections from the light sources indicated that the tear film was well formed in *Smad4^kera-cko^* and the littermate control. However, unlike the control group with the round reflection of two light sources, altered and irregular reflections from light sources were found in the *Smad4^kera-cko^* mice, which indicates that the ocular surface has an irregular shape (Figure 3C,D). The altered ocular surface can cause decreased VP due to the altered refraction angles and the existence of multiple Focal Points other than Principal Focal Point. Furthermore, the altered dimensions from the two sources in the *Smad4^kera-cko^* indicate substantial cornea expansion compared to the *Smad4^Ctrl^* cornea (Figure 3E–L).

### 3.4. Histopathological Changes

Our previous findings in diagnosis level including cornea thinning and a transparent cornea with normal IOP in *Tgfbr2^kera-cko^* were recapitulated in this current study using *Smad4^kera-cko^*. Whether the histological changes including the corneal epithelium thickening, corneal stroma thinning, and corneal epithelial basal cell morphology also mimic the *Tgfbr2^kera-cko^* findings needs further confirmation. To have a better understanding of the tissue and cellular level and see whether the findings in OCT, iCare Tonolab and stereo microscope in *Smad4^kera-cko^* followed the same rationale found in *Tgfbr2^kera-cko^*, we first checked the SMAD4 expression in the *Smad4^kera-cko^* mice to see whether the *Smad4* was a conditional knockout in the keratocyte and then we examined the histopathological changes and cell behavior including the histology, collagen formation, proliferation and differentiation marker. Like the littermate controls, the *Smad4^kera-cko^* mice were able to develop normal eye appendages. The SMAD4 expression can be found in the corneal epithelium and corneal stroma in *Smad4^Ctrl^* close to the nuclei (Figure 4A–C, Epi area, red arrow for SMAD4 in the Str and yellow arrow for nuclei in the Str) and the corneal epithelium in the *Smad4^kera-cko^* (Figure 4D–F, Epi area). However, the SMAD4 expression could not be detected in the corneal stroma of *Smad4^kera-cko^* (Figure 4D Str area). The SMAD4 immunostaining indicated that the *Smad4* was deleted successfully in the keratocytes.

#### 3.4.1. Corneal Epithelium

The corneal epithelium consists of five to seven cell layers including a basal cell layer, wing cell layers, and squamous cell layers, unlike the corneal stroma, and has the major role of providing the diopter for human VA. It also serves as the outermost protective component, playing multiple functions including migration, proliferation, forming the bowman’s membrane in humans and the basement membrane in mice to guarantee the function of the corneal stroma and further serves as a pool for ocular surface compensation to uneven corneal stroma thinning, which is a characteristic used in LASIK surgery [20,21,22]. Our previous study using *Tgfbr2^kera-cko^* had shown that basal epithelial cells exhibited enlargement and irregular arrangement when compared to the control group, which recapitulates the clinical findings in CE patients. Similar to *Tgfbr2^kera-cko^*, in *Smad4^kera-cko^*, our H&E histology staining confirmed the finding in OCT that the cornea is substantially thinner in *Smad4^kera-cko^* as compared to that in the littermate control (Figure 5A–D). Furthermore, in detail information, the corneal epithelium layer in *Smad4^kera-cko^* is substantially thicker than the control group. The H&E histological staining illustrated that the thinner corneas found in the OCT were contributed by the thinner corneal stroma, which is consistent with *Tgfbr2^kera-cko^*.

At the cellular level, compared to the control group, a substantial hypertrophic pathological change was found in *Smad4^kera-cko^* corneal epithelial basal cell. Depending on the basal cell in *Smad4^kera-cko^*, the elongated enlargement was 1.5 to 2 times compared to those in *Smad4^Ctrl^*. The pathological changes of hypertrophy and enhanced stratification were also detected in *Smad4^kera-cko^* wing cell and superficial cell, which indicates a hyperplasia pathological change. The similarities shared between *Tgfbr2^kera-cko^* and *Smad4^kera-cko^* suggested that compromised SMAD4 signaling participated in the generation of the cornea thinning phenotype found in the *Tgfbr2^kera-cko^* mouse line.

Morphological changes in the epithelial basal cells prompted us to examine epithelial cell differentiation in response to the original *Smad4* ablation from the keratocytes. We found that the K12 in the corneal epithelium remained unchanged in *Smad4^kera-cko^* as compared with those in their *Smad4^Ctrl^* littermates (Figure 6A,B). We further examined the proliferation using PCNA immunostaining to see whether the enhanced proliferation found in the *Tgfbr2^kera-cko^* epithelium could be recapitulated in the *Smad4^kera-cko^* mice and might serve as an explanation for the corneal epithelium thickening. The result of the PCNA immunostaining revealed an enhanced proliferation in the *Smad4^kera-cko^* mice in which the positive immunostaining substantially increased in the corneal epithelium (Figure 6C–I) (*n* = 6, ****, *p* < 0.0001). Unlike the *Smad4^Ctrl^* mice had the positive staining restricted to the basal cell (Figure 6C,E). The *Smad4^kera-cko^* mice exhibited positive staining in the wing cells, which are in the second and third layers in the corneal epithelium (Figure 6F,F′,F″,H,H′,H″). The finding of corneal epithelium proliferation further recapitulated the phenotype found in the *Tgfbr2^kera-cko^* in the aspect of cell functional assay.

Furthermore, in this study, we found that the pathogenesis of overall thinning of the cornea accompanied with thinner cornea stroma and thicker cornea epithelium only happened at the postnatal stage (Figure 7A–D).

#### 3.4.2. Corneal Stroma

The corneal stroma is a highly organized collagenous matrix consisting of multiple collagenous lamellae and keratocytes. SMAD4-dependent signaling in keratocytes is known for administering collagen formation and regulation [7]. Our previous CE studies using *Tgfbr2^kera-cko^* revealed a significant reduction in collagen type 1 formation and a significant decrease in keratocyte numbers in the corneal stroma [10]. In this current study, H&E histological staining of *Smad4^kera-cko^* revealed pathological thinning of the corneal stroma, which recapitulates the finding in *Tgfbr2^kera-cko^*. To confirm that the corneal stroma thinning in the *Smad4^kera-cko^* is related to the formation of collagen and follows the rationale in *Tgfbr2^kera-cko^*, we performed Masson’s Trichrome staining in the *Tgfbr2^kera-cko^*, *Tgfbr2^Ctrl^*, *Smad4^kera-cko^* and *Smad4^Ctrl^* mouse lines to visualize the collagen formation and distribution in the corneal stroma. Masson’s Trichrome staining revealed a substantial reduction in collagen in the corneal stroma in *Smad4^kera-cko^* (Figure 8B) and *Tgfbr2^kera-cko^* (Figure 8D) compared to their control groups, respectively (Figure 8A,C). The findings in *Smad4^kera-cko^* and *Tgfbr2^kera-cko^* implied that a compromised collagen formation in *Smad4^kera-cko^* and *Tgfbr2^kera-cko^*, which further correlated a compromised SMAD4-dependent signaling to the *Tgfbr2*-related CE-like phenotype in the aspect of collagen formation in the corneal stroma.

However, unlike *Tgfbr2^kera-cko^*, the phenotype of uneven distribution and clusters of keratocytes in the corneal stroma cannot be recapitulated in *Smad4^kera-cko^* mice, indicating the possibility that the pathogenesis and the pathophysiology of keratocyte, although they exhibit the same phenotype in OCT, might differ (Figure 5A–D,D′,D″,D‴, Figure 8A–D). Considering the thinning of the corneal stroma, the total stromal cell density was found to be significantly higher in *Smad4^kera-cko^* compared to *Smad4^Ctrl^* (Figure 8A–D,K) (*n* = 6, ****, *p* < 0.0001).

To further confirm the finding of the Masson’s Trichrome staining in *Smad4^kera-cko^*, we performed immunostaining of collagen type 1 (Col1a1) in *Smad4^kera-cko^* and *Smad4^Ctrl^*. Consistent with the finding of Masson’s Trichrome and our previous paper using *Tgfbr2^kera-cko^*, the collagen type 1 formation was substantially lower in the *Smad4^kera-cko^* (Figure 8H–J) compared to that of *Smad4^Ctrl^* (Figure 8E–G). Moreover, a noticeable discontinued Col1a1 staining was found in the *Smad4^kera-cko^* in the posterior and anterior corneal stroma (Figure 8H red *).

### 3.5. Keratocytes Attache to Descemet’s Membrane in the Absence of SMAD4-Dependent Signaling

In our previous study, eye-rubbing served as the representative of the environmental factors examined in our previous animal model of *Tgfbr2^kera-cko^*. The finding of the eye-rubbing experiment indicated the importance and the significance of the involvement of environmental factors in the pathogenesis of the CE-like phenotype by showing the similarities between the classic signs found in acute/advanced CE, including keratoconus, keratoglobus, and a PMD called Acute Corneal Hydrops (ACH), after the disorder was triggered by the genetic defect of TGFBR2 in the corneal stroma. The ACH sign is due to the rupture of Descemet’s membrane, which indicates that Descemet’s membrane suffered due to the pathological thinning of the corneal stroma, and the homeostasis, together with the structure, could not be maintained in our previous animal model.

In this study, to test the homeostasis and the health of the Descemet’s membrane in the aspect of functional analysis in our current animal model, we applied the eye-rubbing experiment using the same strategy. *Smad4^kera-cko^* mice, unlike the *Tgfbr2^kera-cko^* mice, exhibited Acute Corneal Hydrops signs, although they also exhibited cornea thinning with a thinner corneal stroma, survived from the eye-rubbing experiment, which indicates that the Descemet’s membrane structure should remain intact. To confirm our hypothesis, we performed TEM to check the Descemet’s membrane and keratocyte situation. The TEM data revealed an intact and healthy Descemet’s membrane across the whole cornea and a normal keratocyte appearance in *Smad4^kera-cko^,* although they also exhibited pathological thinning in the corneal stroma (Figure 9A,B Str, I, 36.55 ± 0.53 μm vs. 106.72 ± 0.90 μm, *n* = 6, ****, *p* < 0.0001) using Toluidine blue staining (Figure 9A–H, Des in red dash line). However, a noticeable attachment of the keratocytes to the Descemet’s membrane was found in *Smad4^kera-cko^* (Figure 10A–D, red arrow). Furthermore, the keratocyte attached to the Descemet’s membrane exhibited irregular morphology (Figure 10A–D, Kera area) compared to the normal keratocytes, which usually have broad, fan-shaped lamellipodium (Figure 9C,D).

The different findings in the keratocytes and the Descemet’s membrane in the current animal model *Smad4^kera-cko^* might serve as an explanation for the clustering and disappearing of the keratocytes found in *Tgfbr2^kera-cko^*. Furthermore, the organization of the lamellae was maintained, and the diameters of collagen fibrils remained the same compared to the control group (Figure 11A–D,F), which indicates that although the SMAD4-dependent signaling pathway can contribute to the stroma thinning, the micro-environment in the corneal stroma remains relatively healthy and the structure is still in the proper orientation and provides enough stiffness against the eye-rubbing experiment. However, although the micro-environment is relatively healthy, the density of the collagen fiber showed a significant decrease in the stroma (Figure 11E).

### 3.6. OCE Revealed a Softer Cornea in Smad4^kera-cko^

To elucidate whether the corneas are softer in the *Smad4^kera-cko^* compared to those in *Smad4^Ctrl^*, we performed an OCE to examine the Wave Speed, Young’s Modulus, and the Viscosity. When we compared the *Smad4^kera-cko^* to *Smad4^Ctrl^* at 10 mmHg IOP (the normal mouse IOP), significant decreases were found in the Wave Speed (3.39 ± 0.82 vs. 2.78 ± 0.40 m/s), Young’s Modulus (2.21 ± 0.92 vs. 1.58 ± 0.59 MPa), and the Viscosity (2.06 ± 0.52 vs. 1.15 ± 0.29 Pa.s.), which indicates the corneas in *Smad4^kera-cko^* are softer than those in *Smad4^Ctrl^*.

## 4. Discussion

In clinics, although the treatments for CE patients, including Collagen Cross-Linking (CXL) and Intrastromal Corneal Ring Segments implantation (ICRS), can stop the progression of CE via increasing corneal biomechanical resistance and correct CE by shortening the cone length, respectively, in advanced CE, cornea transplant surgery is not avoidable for treating the extreme pathological cornea thinning and uncontrollable vision loss [23,24,25,26,27,28]. Herein, further elucidating the etiology and pathogenesis for improving the quality of medical treatment in the aspect of molecular biology, specifically the intracellular signaling pathway, becomes necessary to prevent and intervene in CE progression in clinics.

Our previous study indicated that the deficiency of TGFBR2 in the corneal stroma can result in cornea thinning characterized by pathological thinning of the corneal stroma accompanied by thickening of the corneal epithelium [10]. In this current study, we wanted to further investigate whether the SMAD4-dependent signaling is involved in the generation of the CE-like phenotype in the aspect of Loss-of Function, aiming to elucidate the role of collagen formation in our previous findings. Our data revealed certain similarities between *Smad4^kera-cko^* and *Tgfbr2^kera-cko^* at the diagnosis level, tissue level, and cellular level. Considering that the CCT found in the OCT showed no difference between *Smad4^kera-cko^* and *Tgfbr2^kera-cko^* and the ultra-pathological thinning found in the *Smad4^kera-cko^* toluidine blue staining (Figure 9A,B), a conclusion of the involvement of a compromised SMAD4-dependent signaling pathway in the pathogenesis of TGFBR2-deficiency related to the CE-like phenotype could be made.

However, the difference between the results of the eye-rubbing experiment and intact of Descemet’s membrane, as well as the healthy keratocyte in our TEM of the *Smad4^kera-cko^* mice, suggested the existence of another rationale and molecular mechanism behind the similarities in the diagnosis levels contributing to the Descemet’s membrane weakening and keratocyte surviving, which further explains why in the clinics, the progressions in CE patients vary with the similarities in the signs and symptoms when the event is triggered by genetic defects. Elucidating the molecular mechanism can give us insight to further classify the CE disorders and provide better targets for clinicians to intervene.

### 4.1. A Compromised SMAD4-Dependent Signaling Might Contributed to the TGFBR2-Deficiency Related CE-like Phenotype

The SMAD4-dependent signaling pathway serves as the canonical pathway of TGFB signaling and it plays an important role in the formation of collagen type 1, which is the major component in the corneal stroma. Our result from the Masson’s Trichrome staining helped us to visualize the collagen and revealed a substantial reduction in collagen formation in the corneal stroma of *Smad4^kera-cko^* and *Tgfbr2^kera-cko^*. Moreover, the collagen type 1 (Col1a1) immunostaining in the *Smad4^kera-cko^* further confirms the findings from Masson’s Trichrome and revealed a less abundant and discontinued Col1a1 immunostaining compared to that in the control group. The findings of the discontinued collagen type 1 staining also recapitulate the findings in our previous study in *Tgfbr2^kera-cko^*.

Other than the collagen formation, the TEM and OCE revealed a decrease in the collagen density and a decrease in the corneal stiffness, respectively. Considering that the iCare Tonolab did not show a decreased reading in the IOP in *Smad4^kera-cko^*, the real IOP under a softer cornea might be higher. An increased IOP and the decreased collagen density in the *Smad4^kera-cko^* might serve as the mechanisms of a CE-like phenotype generation at postnatal corneal development. Furthermore, in this study, *Smad4^kera-cko^* mice shared many similarities with *Tgfbr2^kera-cko^* mice in diagnosis, tissue, and cellular level, including cornea thinning companies with corneal epithelium thickening and pathological thinning of the corneal stroma, hypertrophy and hyperplasia in corneal epithelial cell and transparent cornea with normal IOP reading. Taking these findings together, the involvement of a compromised function of SMAD4-dependent signaling in the generation of the TGFBR2 deficiency-related CE-like phenotype is convincing.

### 4.2. SMAD4-Independent Signaling Might Play an Important Role in Keratocyte Surviving

In our previous study, the keratocyte in *Tgfbr2^kera-cko^* suffered from pathological changes characterized by a vacuole appearance [10]. However, the keratocyte in *Smad4^kera-cko^* did not exhibit the same pathological appearance, which indicates that the pathological changes in the keratocyte found in the *Tgfbr2^kera-cko^* mice were not directly generated by the SMAD4-dependent signaling pathway in the aspect of Loss-of-Function. TGFBR2 serves as the initiation component control of many downstream signaling pathways including the SMAD4-dependent pathway. Other than SMAD4-dependent signaling, SMAD4-independent is the other group of pathways triggering different signaling, including MAPK/ERK, Tak1 (Mitogen-activated protein kinase kinase kinase 7, MAP3K7), JNK (c-Jun N-terminal kinases) and PI3K [29,30,31,32]. While SMAD4-dependent signaling is in charge of collagen formation in the corneal stroma, SMAD4-independent signaling, on the other hand, plays an important role in the cell cycle, immune response, inflammation, cell growth, proliferation, differentiation, motility, survival, ER-stress regulation, and intracellular trafficking, respectively. Although the similarities shared among *Smad4^kera-cko^*, *Tgfbr2^kera-cko,^* and CE patients indicate the involvement of SMAD4-dependent signaling in the CE-like pathogenesis, the pathological appearance in the keratocyte of *Tgfbr2^kera-cko^* mice might result from the SMAD4-independent signaling pathway. In other words, SMAD4-independent signaling might play an important role in the keratocyte survival.

### 4.3. Pathological Changes in Keratocyte Might Serve as a Fundamental Background in the Pathogenesis of Advanced CE

Acute Corneal Hydrops, an uncommon complication of CE, involves corneal edema due to a sudden rupture in the Descemet’s membrane and can cause visual defects [33,34,35]. The occurrence of corneal hydrops is a sign of advanced CE, including keratoconus, keratoglobus, and PMD. Our previous study using *Tgfbr2^kera-cko^* mice showed that the corneal hydrops phenotype can be induced by eye-rubbing. However, the corneal hydrops phenotype cannot be recapitulated in this current study using *Smad4^kera-cko^*, which indicates the different conditions in the Descemet’s membrane between *Tgfbr2^kera-cko^* mice and *Smad4^kera-cko^*. Considering that SMAD4-dependent signaling is responsible for the collagen formation and the findings of the keratocytes attachments to Descemet’s membrane due to the ultra-pathological thinning of corneal stroma in *Smad4^kera-cko^*, the pathological changes in keratocytes and the elimination/absence of posterior keratocytes found in *Tgfbr2^kera-cko^* might serve as the fundamental background of ACH in advanced CE. The rationale could be that, due to the pathological thinning of the corneal stroma, some of the keratocytes start to attach to Descemet’s membrane and the following pathological changes in keratocytes might cause the cell death, which further impacts the health and structure of the Descemet’s membrane structure. Although the event is transient and hard to monitor in TEM, such a negative effect can provide a background for ACH happening when environmental factors such as eye-rubbing are involved.

## 5. Conclusions

In this study, we elucidated the involvement of the SMAD4-dependent signaling pathway in the aspect of Loss-of-Function, in which the corneal thinning phenotype can be recapitulated in *Smad4^kera-cko^*. However, the classic sign of advanced CE, called ACH, was missing in the current study, which indicates that the molecular mechanisms behind the keratocyte pathogenesis in these two animal models are not exactly the same. Furthermore, how the SMAD4-independent signaling is involved in the pathological changes in keratocytes remains unclear and needs further investigation. Although signaling pathways and molecular mechanisms are complicated due to the compensation and the interaction between different signaling pathways, conditional knockout animal models with different effect genes can give us an insight and possible explanation behind the phenotype and observation, which further provide more convincing criteria for clinics to categorize the disease with similar signs and symptoms.

## Figures and Tables

**Figure 1 cells-13-00626-f001:**
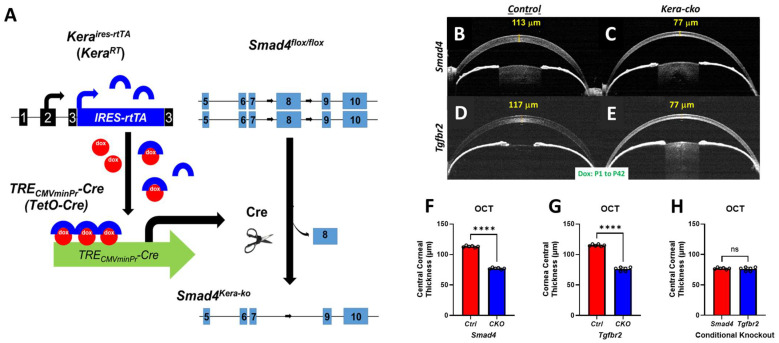
***Smad4^kera-cko^* revealed corneal thinning phenotype similar to those in *Tgfbr2^kera-cko^*.** (**A**) Schematic drawing of *Smad4* deletion in *Kera^rtTA^*; *TetO-Cre*; *Smad4^flox/flox^* mouse line. The rtTA is constitutively synthesized from *Kera^rtTA^* alleles in keratocytes but remains in an inactive form. Upon binding of doxycycline (dox) to rtTA, the dox-rtTA complex transcriptionally activates *Tet-O-Cre* and produces Cre recombinase, which then subsequently splices *loxP* sites and flank the exon 8. (**B**–**G**) OCT revealed thinner corneas in *Smad4^kera-cko^* and *Tgfbr2^kera-cko^* compared to those in the control group respectively at P42 (*n* = 6, ****, *p* < 0.0001). (**C**,**E**,**H**) However, the Central Corneal Thickness of *Smad4^kera-cko^* corneas showed no significant difference compared to those in *Tgfbr2^kera-cko^* (*n* = 6, ns). Abbreviation: ns, not significant.

**Figure 2 cells-13-00626-f002:**
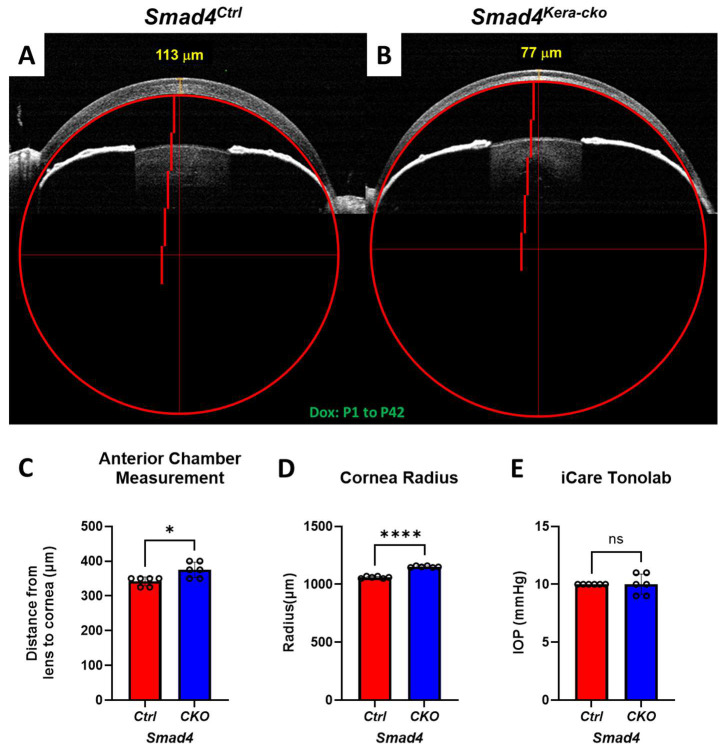
***Smad4^kera-cko^* revealed a larger cornea with enlarged Anterior Chamber under normal IOP reading.** (**A**,**D**) Cornea Radius measurement indicating significance difference between *Smad4^kera-cko^* and *Smad4^Ctrl^* at P42 (*n* = 6, ****, *p* < 0.0001). (**A**–**C**) Anterior chamber measurement using the distance between lens and cornea showed longer distances in *Smad4^kera-cko^* compared to those in *Smad4^Ctrl^* (*n* = 6, *, *p* = 0.0133). (**E**) IOP measurement using iCare Tonolab exhibited no significant difference between *Smad4^kera-cko^* and *Smad4^Ctrl^* (*n* = 6, ns). Abbreviation: ns, not significant.

**Figure 3 cells-13-00626-f003:**
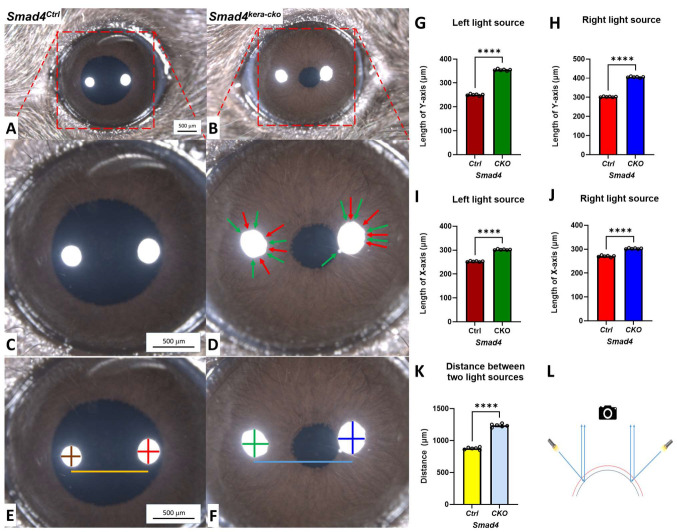
**Stereo microscope indicating expanded corneas with abnormal light reflection in *Smad4^kera-cko^*.** (**A**–**D**) The abnormal reflection (red arrow for the concave and green arrow for the convex) indicates the irregular corneal surface in *Smad4^kera-cko^* compared to those in *Smad4^Ctrl^*, which can cause decreased Visual Precision (VP) due to the irregular light refractions. (**E**–**K**) Corneal expanding measurement showed significant all-direction expansions in *Smad4^kera-cko^* by measuring the distance between two light sources and the *X*-axis and *Y*-axis within the light sources (the colors in the chart are consistence with the colors in the measuring lines, *n* = 6, ****, *p* < 0.0001). (**L**) Schematic drawing of the measuring rationale (the red line indicates the ectatic cornea with early reflection of the light paths, which results in a larger distance between two reflections of light sources, and the black cornea represents the normal cornea with a shorter distance between two light reflections).

**Figure 4 cells-13-00626-f004:**
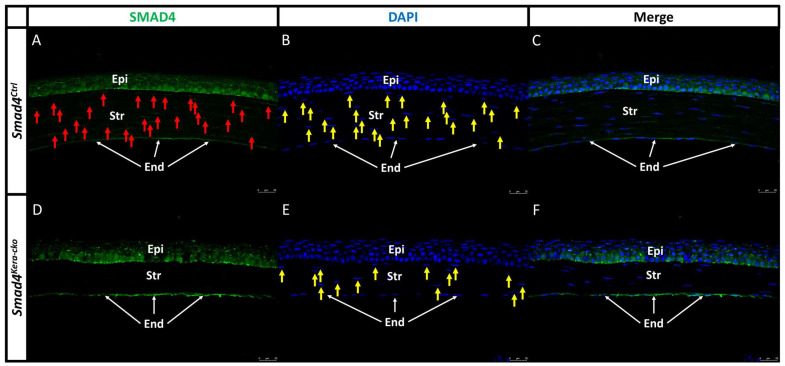
**SMAD4 Immunofluorescent staining in *Smad4^Ctrl^* and *Smad4^kera-cko^*.** (**A**,**C**,**D**,**F**) SMAD4 immunostaining (green) showed that the SMAD4 expression can be detected in both *Smad4^Ctrl^* and *Smad4^kera-cko^* corneal epithelium (Epi) (**A**,**D**) but can only be detected in *Smad4^Ctrl^* corneal stroma (**A**, Str, red arrow) and cannot be detected in *Smad4^kera-cko^* stroma (**D**, Str) due to the conditional knockout of *Smad4* in keratocyte. (**B**,**D**) DAPI immunostaining (blue) revealed the location of the nucleus of keratocyte in the corneal stroma (**B**,**E**, Str, yellow arrow) and the locations are mostly consistent with the positive immunostaining of SMAD4 in the *Smad4^Ctrl^* (**A**–**C**). The reason why some of the SMAD4 immunostaining cannot have the corresponding DAPI immunostaining in stroma and epithelium is that the DAPI only shows where the nucleus is at but the SMAD4 usually stays in the cytoplasm. Considering that the thickness of our sectioning is 5 μm (the dimension of a normal corneal keratocytes is usually over 5 μm and the base diameter of normal corneal epithelial basal cell is around 4–8 μm), the positive SMAD4 immunostaining should be around, overlap or close to the nucleus in both corneal stroma and epithelium. However, if the sectioning only cuts through the cytoplasm and barely touches the nucleus, then we can only have the positive SMAD4 immunostaining without DAPI, and vice versa. Abbreviations: Epi, corneal epithelium; Str, corneal stroma; End, corneal endothelium.

**Figure 5 cells-13-00626-f005:**
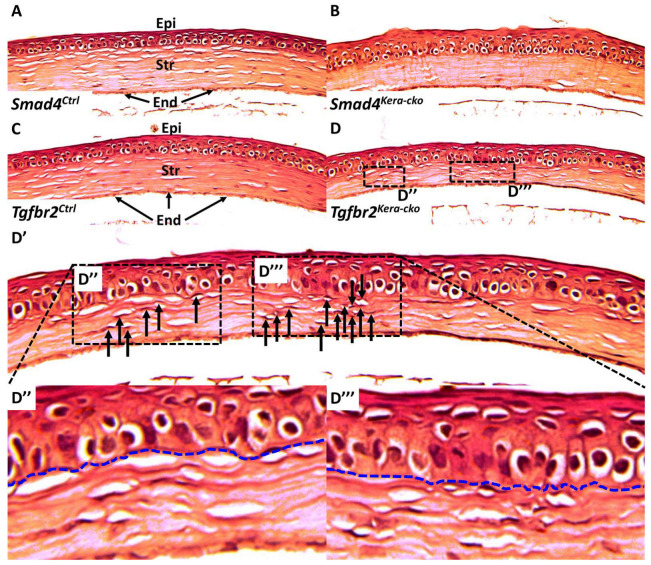
**H&E staining in *Smad4* and *Tgfbr2* mouse lines.** (**A**,**C**) HE-staining revealed the normal cornea in control groups including the normal ratio of corneal epithelium (Epi) and stroma (Str) and the normal distribution of keratocytes in the corneal stroma. (**B**,**D**) Both *Smad4^kera-cko^* and *Tgfbr2^kera-cko^* exhibited overall corneal thinning with thinner corneal stroma and thicker corneal epithelium characterized by hypertrophy in the basal cell. (**A**–**D**) The distribution of keratocytes in the *Smad4^kera-cko^* remains normal and even compared to those in *Smad4^Ctrl^* (**A**,**B**). However, the cluster and gathering of keratocytes can be noticed in the *Tgfbr2^kera-cko^* (**D**, **D′** dashed square) compared to those in *Tgfbr2^Ctrl^*, *Smad4^Ctrl,^* and *Smad4^kera-cko^*. The uneven distribution of keratocytes in *Tgfbr2^kera-cko^* recapitulates the findings in our previous study. Furthermore, the irregular interfaces between the corneal epithelium and corneal stroma can be found in the *Tgfbr2^kera-cko^* and correspond to the keratocyte gathering (**D′**,**D″**,**D‴**, blue dash). (**D‴**) The severe gathering of the keratocytes (12 keratocytes (black arrows) can be found on the right side of **D‴** and 6 keratocytes (black arrow) in the left side of **D″**) can also be found to correspond to the concave inward to the interface between corneal epithelium and corneal stroma (**D″**,**D‴**). Compared to *Tgfbr2^kera-cko^*, the interfaces are smooth in the *Smad4^Ctrl^*, *Tgfbr2^Ctrl,^* and *Smad4^kera-cko^* (**A**–**C**). Abbreviations: Epi, epithelium; Str, stroma; End, endothelium.

**Figure 6 cells-13-00626-f006:**
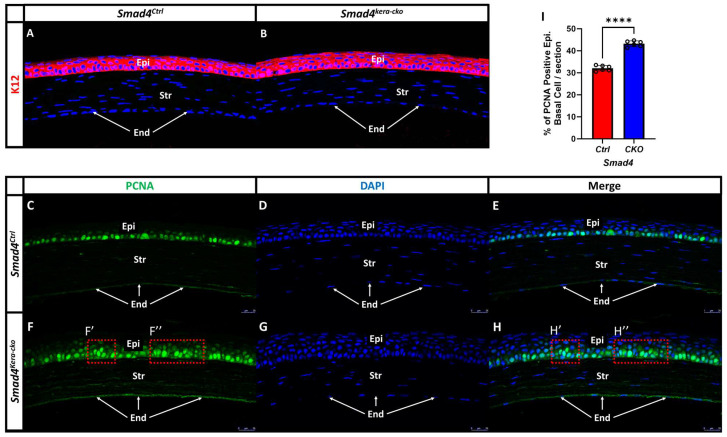
**PCNA and K12 Immunofluorescent staining in *Smad4^Ctrl^* and *Smad4^kera-cko^*.** (**A**,**B**) Corneal epithelial specific marker K12 remains unchanged in *Smad4^kera-cko^* compared to *Smad4^Ctrl^*. (**C**,**E**,**F**,**H**) Proliferating Cell Nuclear Antigen (PCNA) immunostaining (green) showed that the basal cell of corneal epithelium (Epi) underwent DNA replication and proliferation in both *Smad4^kera-cko^* and *Smad4^Ctrl^* (**C**,**F**). However, in the *Smad4^kera-cko^*, a significant increase of positive PCNA immunostaining can be noticed (43.17 ± 0.54%) compared to *Smad4^Ctrl^* (32.07 ± 0.50%) (*n* = 6, ****, *p* < 0.0001) (**I**) and some of the positive immunostaining can be found in the second and third layer of corneal epithelium which is where the Wing Cell at (**F′**,**F″**,**H′**,**H″**). (**D**,**G**) DAPI immunostaining (blue) revealed the location of the nucleus of epithelial cell in the corneal epithelium and the locations are mostly consistent with the positive immunostaining of PCNA in the *Smad4^Ctrl^* and *Smad4^kera-cko^* (**C**–**H**). The reason why some of the PCNA immunostaining is weaker and cannot have the 100% corresponding DAPI immunostaining in the epithelium is that the thickness of our sectioning is 5 μm (the base diameter of normal corneal epithelial basal cell is around 4–8 μm but the corneal epithelial cell in *Smad4^kera-cko^* showed hypertrophy pathological changes which can cause a substantial increase in diameter of the epithelial basal cell and can over 5 μm), the positive PCNA immunostaining sometimes is overlap or close to another nucleus. However, if the sectioning of the particular area in the figure only contains one epithelial cell and barely touches multiple cell nuclei, then we can have a positive PCNA immunostaining with sharp nucleus boundaries and the PCNA immunostaining can overlap with nucleus DAPI immunostaining staining perfectly, and vice versa. Abbreviations: Epi, corneal epithelium; Str, corneal stroma; End, corneal endothelium.

**Figure 7 cells-13-00626-f007:**
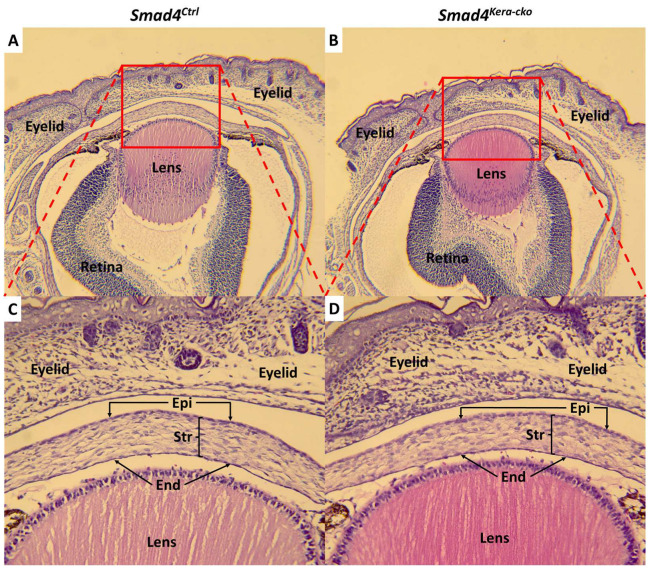
**H&E staining at postnatal day 1 in *Smad4* mouse line.** The dox was given at embryonic day 0 and the eyeballs were collected at postnatal day 1 (between postnatal 0 h to 24 h) (**A**,**B**) HE-staining revealed the normal development of eyeball including the good formation of the corneal epithelium (Epi), endothelium (End) and stroma (Str) (**C**,**D**) and the normal distribution of keratocyte in corneal stroma in both *Smad4^Ctrl^* and *Smad4^kera-cko^*. Abbreviations: Epi, epithelium; Str, stroma; End, endothelium.

**Figure 8 cells-13-00626-f008:**
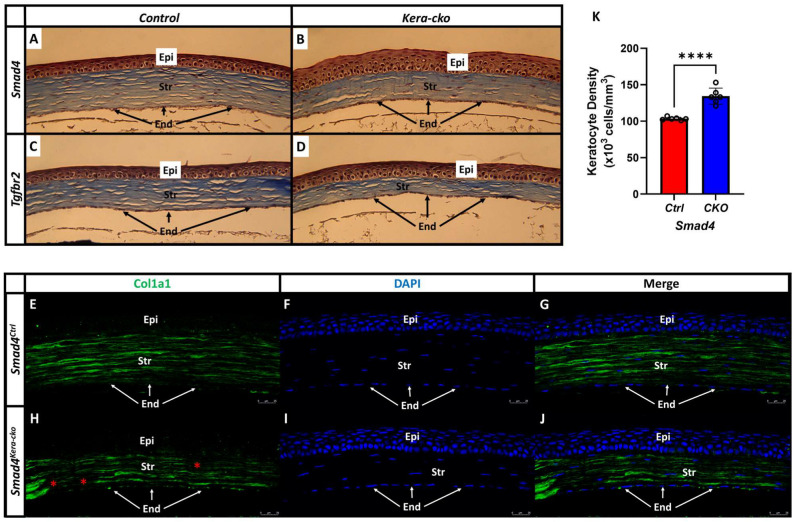
**Collagen type 1 formation is less abundant in *Smad4^kera-cko^* and *Tgfbr2^kera-cko^*.** (**A**,**C**) Trichrome staining revealed a normal cornea in control groups including the normal ratio between corneal epithelium (Epi) and stroma (Str) and the normal distribution of keratocytes in the corneal stroma. (**B**,**D**) Both *Smad4^kera-cko^* and *Tgfbr2^kera-cko^* exhibited overall corneal thinning with thinner corneal stroma and thicker corneal epithelium characterized with hypertrophy in the epithelial basal cell compared to their control groups respectively. (**A**–**D**) The collagens were better visualized in the Masson’s Trichrome staining (blue) in corneal stroma. (**B**,**D**) Compared to the control groups, both *Smad4^kera-cko^* and *Tgfbr2^kera-cko^* exhibited a substantial decrease of the collagen staining (blue), which indicates a substantial decrease of the collagen formation in the stroma of *Smad4^kera-cko^* and *Tgfbr2^kera-cko^*. (**A**,**B**,**K**) Furthermore, unlike the disappearing and uneven distribution found in the *Tgfbr2^kera-cko^* keratocyte, the distribution of the keratocytes in *Smad4^kera-cko^* remains normal. Due to the thinning of the corneal stroma, the density of the corneal keratocytes was found increased (103.33 ± 0.84 vs. 134.33 ± 4.49, *n* = 6, ****, *p* < 0.0001). (**E**,**G**,**H**,**J**) Collagen type 1 immunostaining (Col1a1, green) revealed a substantial decrease of the collagen type 1 in the *Smad4^kera-cko^* compared to those in the *Smad4^Ctrl^*. (**H**) In *Smad4^kera-cko^*, a discontinued Col1a1 staining can be noticed in the posterior stroma and anterior stroma (red *). (**F**,**I**) DAPI immunostaining (blue) revealed the location of the nucleus of epithelial cells and keratocyte. Abbreviations: Epi, corneal epithelium; Str, corneal stroma; End, corneal endothelium.

**Figure 9 cells-13-00626-f009:**
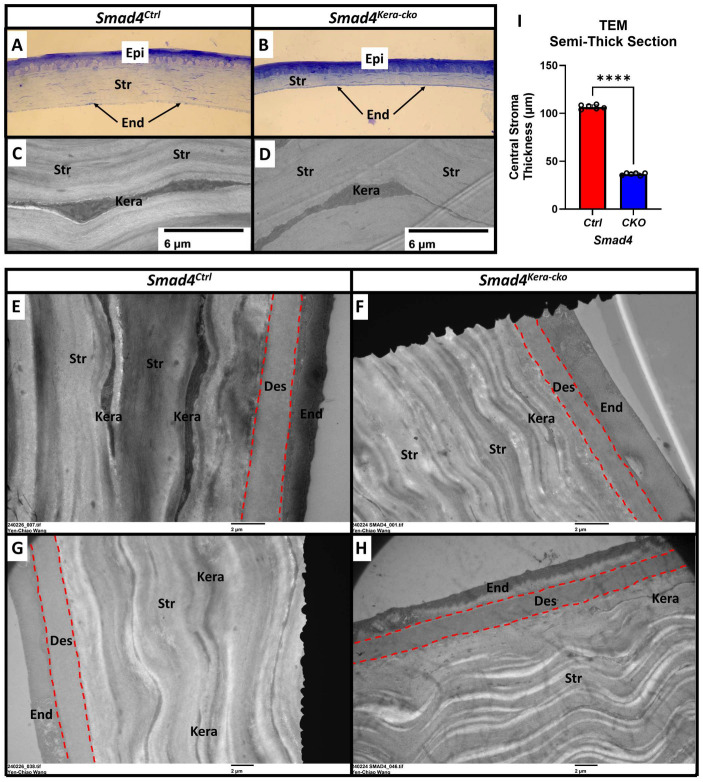
**TEM revealed a normal keratocyte appearance with an intact Descemet’s membrane.** (**A**,**B**,**I**) Representative Toluidine Blue-Stained Semi-Thick Scout Section revealed a pathological thinning in the corneal stroma in *Smad4^kera-cko^* (36.55 ± 0.53 μm) compared to those in *Smad4^Ctrl^* (106.72 ± 0.90 μm) (*n* = 6, ****, *p* < 0.0001) and both the interfaces between corneal epithelium and corneal stroma are smooth. (**C**,**D**) TEM revealed a normal appearance in keratocytes in both *Smad4^Ctrl^* and *Smad4^kera-cko^*. (**E**–**H**) The Descemet’s membrane is intact and showed no defect in the *Smad4^kera-cko^*, neither a significant difference compared to those in *Smad4^Ctrl^* (Red dash lines). Abbreviations: Epi, epithelium; Str, stroma; End, endothelium; Kera, keratocyte; Des, Descemet’s membrane.

**Figure 10 cells-13-00626-f010:**
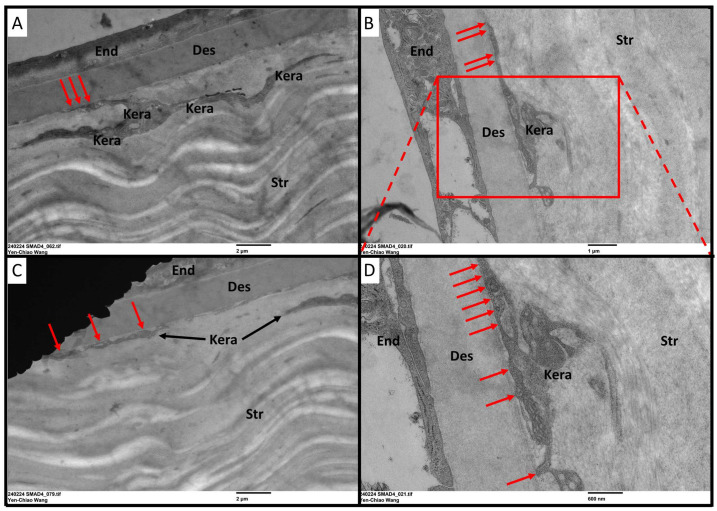
**TEM revealed a keratocyte attachment to the Descemet’s membrane in the *Smad4^kera-cko^* mouse line.** (**A**–**D**) Unlike the normal corneal stroma usually have the abundant collagen fibrils between the posterior keratocytes and the Descemet’s membrane, due to the decrease of the collagen type 1 in the stroma, the keratocytes in *Smad4^kera-cko^* were noticed to be attached to the Descemet’s membrane (red arrows) in the posterior stroma. An abnormal morphology can also be noticed compared to the normal keratocytes. Abbreviations: Str, stroma; End, endothelium; Kera, keratocyte; Des, Descemet’s membrane.

**Figure 11 cells-13-00626-f011:**
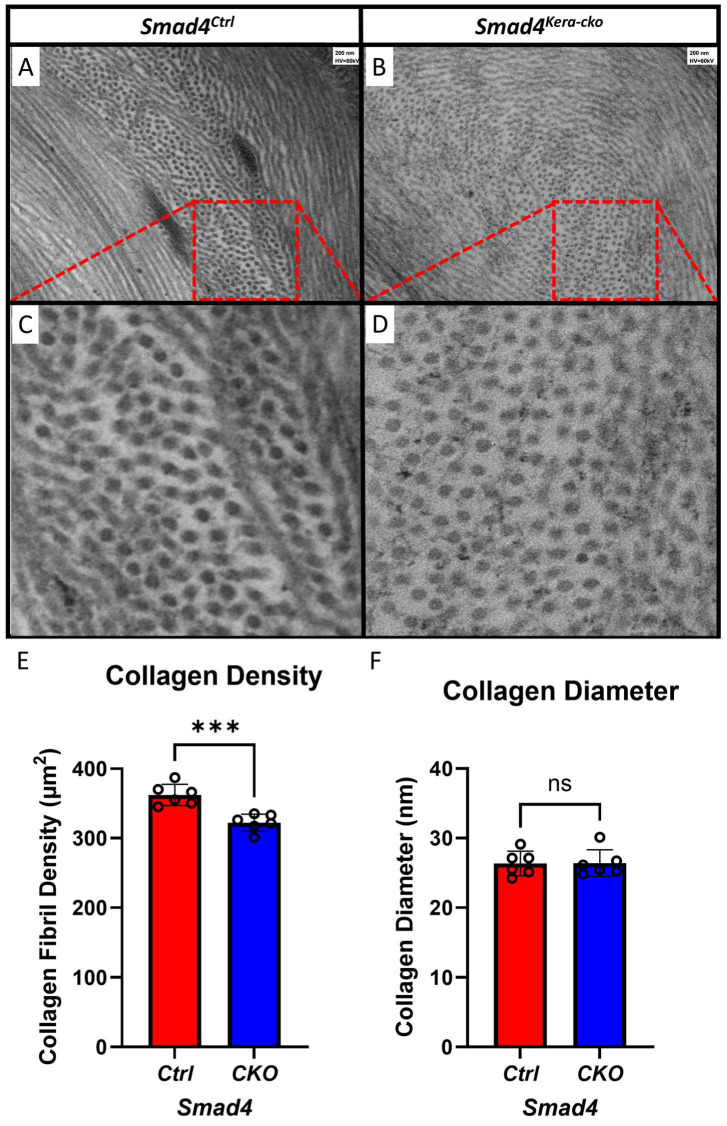
**TEM revealed a decreased collagen density with normal collagen diameter in *Smad4^kera-cko^*.** (**A**,**B**) TEM revealed the organization of collagen in different orientation including vertical and parallel to the sectioning surface of both the *Smad4^Ctrl^* and *Smad4^kera-cko^*. (**C**,**D**) For those collagens that is vertical to the sectioning surface, we perform the collagen density calculation and collagen diameter measuring. (**E**) Collagen density is decreased in the *Smad4^kera-cko^* compared to those in the *Smad4^Ctrl^* (362.17 ± 6.29 vs. 322.17 ± 5.08, *n* = 6, *p* = 0.0007, ***). (**F**) Collagen diameter measurement showed no significant difference between *Smad4^kera-cko^* and *Smad4^Ctrl^* (*n* = 6, *p* = 0.9758, ns). Abbreviation: ns, not significant.

## Data Availability

Dataset available on request from the authors.

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
