# Peer review of "SMAD4-Dependent Signaling Pathway Involves in the Pathogenesis of TGFBR2-Related CE-like Phenotype"

_cells, 2024, doi:10.3390/cells13070626_

Round 1

Reviewer 1 Report

Comments and Suggestions for Authors

In this manuscript, the authors construct a conditional knockout of SMAD4 in mice and detect corneal phenotype associated with cornea ectasia. While this study is interesting and the authors analyzed a large number of parameters, there are several important questions.

1.     One of the main concerns is that the study was mainly descriptive and did not address the underlying mechanisms between the deletion of SMAD4 and cornea ectasia. Although the authors listed a large amount of data showing the phenomenon of cornea ectasia, no explanation was given as to why the absence of SMAD4 caused cornea ectasia.

2.     Another major concern is that the results do not fully support the conclusions. How can the authors demonstrate that SMAD4-dependent signaling pathway is involved in the pathogenesis of CE-like phenotype observed in Tgfbr2kera-cko? Although the major function of SMAD4 is in response to TGF-β signaling, there is no direct evidence to suggest the relation of TGF-β and SMAD4 in this study. Moreover, there were some differences in cornea between SMAD4 and TGFBR2 knockout mice.

3.     In Fig.1, the authors compare the thickness of the cornea at P42. Why did the authors choose this time point? Similar questions are shown in Fig.2-4.

4.     In Fig.2.C and D, the images are too blurry to see.

5.     In Fig.2.F, there is no bar value. How many mice were used for IOP measurement?

6.     The methods section to be consistent with the information given for each method. In some cases there is almost no detail and some experiments are lacking.

Comments on the Quality of English Language Please revise English grammar.

Author Response

In this manuscript, the authors construct a conditional knockout of SMAD4 in mice and detect corneal phenotype associated with cornea ectasia. While this study is interesting and the authors analyzed a large number of parameters, there are several important questions.

  1. One of the main concerns is that the study was mainly descriptive and did not address the underlying mechanisms between the deletion of SMAD4 and cornea ectasia. Although the authors listed a large amount of data showing the phenomenon of cornea ectasia, no explanation was given as to why the absence of SMAD4 caused cornea ectasia.

We appreciate reviewer’s comment and revise the manuscript accordingly by performing the TEM aiming the Descemet’s membrane condition including the finding of the keratocytes attachment to the Descemet’s membrane. The OCE data was also added to elucidate the possible mechanism. We further adding the following statement in discussion. “The TEM and OCE revealed a decrease of the collagen density and a decrease of the corneal stiffness respectively. Considering that the iCare Tonolab did not show a decreased reading of IOP in Smad4kera-cko, the real IOP under a softer cornea might be higher. An increased IOP and the decreased collagen density which cause the softer cornea in the Smad4kera-cko might serve as the mechanism of a CE-like phenotype at postnatal corneal development.”.

  1. Another major concern is that the results do not fully support the conclusions. How can the authors demonstrate that SMAD4-dependent signaling pathway is involved in the pathogenesis of CE-like phenotype observed in Tgfbr2kera-cko? Although the major function of SMAD4 is in response to TGF-β signaling, there is no direct evidence to suggest the relation of TGF-β and SMAD4 in this study. Moreover, there were some differences in cornea between SMAD4 and TGFBR2 knockout mice.

We thank reviewer’s valuable comment and the insight of consideration. The major function of SMAD4-dependent signaling is the collagen type-1 formation, which is the major component of the corneal stroma. Our previous study using Tgfbr2kera-cko was revealed with less abundant of collagen type 1 in the corneal stroma which served as one of the mechanisms of the corneal stroma thinning. However, in the previous article, we do not know whether such phenotype is due to the decrease of the collagen formation or an enhance of collagen degradation. To solidate the involvement of SMAD4-dependent signaling and answer reviewer’s comment, we add the experiments of collagen type 1 immunostaining in Smad4kera-cko and Trichrome stain in both Smad4kera-cko and Tgfbr2kera-cko. The result indicated that the stroma thinning was resulted from the collagen formation in both mouse lines and the data can fit each other. On the other hand, we humbly understand that the pathological appearance in the keratocytes cannot be recapitulated in Smad4kera-cko, so in the discussion, we indicated that the pathological changes found in keratocytes might be due to SMAD4-independent signaling. However, the involvement and the contribution of SMAD4-dependent signaling to the corneal stroma thinning still stand based on the tremendous similarities shared between Smad4kera-cko and Tgfbr2kera-cko, which allowing us to make the conclusion of the involvement of SMAD4-signaling.       

  1. In Fig.1, the authors compare the thickness of the cornea at P42. Why did the authors choose this time point? Similar questions are shown in Fig.2-4.

      We appreciate reviewer’s question and comment. In clinics, cornea ectasia usually onset at puberty and undergo slow progression in adult with occasionally acute progression of corneal hydrops. In mouse, P42 is the timing they require the sexual ability, and we choose this timing in mouse for the time-matching purpose as human puberty.

  1. In Fig.2.C and D, the images are too blurry to see.

      We thank the reviewer’s comment. Fig. 2C and Fig. 2D were designed to show the protrusion of the cornea so the focus is on the corneal edge, which makes the fur too blurry. Due to the concern and we think the other information including the measurement of Anterior Chamber and the Cornea Radius are with higher quality, we took away the Fig. 2C and 2D.

  1. In Fig.2.F, there is no bar value. How many mice were used for IOP measurement?

      We appreciate the reviewer’s notice, and we revised it accordingly. The n = 6 and the bar was added.

  1. The methods section to be consistent with the information given for each method. In some cases, there is almost no detail, and some experiments are lacking.

      We thank reviewer’s comment and we have revised the methodology part.

Reviewer 2 Report

Comments and Suggestions for Authors

The purpose of this study is to elucidate whether SMAD4-dependent signaling pathway is involved in the TGFBR2-related CE-like pathogenesis. The results  indicate that SMAD4-dependent signaling pathway involves in the pathogenesis of CE-like phenotype . 

This new findings provided new knowledge for the mechanism of CE disease like keratoconus, which is a blind causing disease.

The manuscript is well writen, the data could support the conclusion.

1.     The main question addressed by the research  is that MAD4-dependent signaling pathway is involved in the TGFBR2-related CE-like pathogenesis.

2.     For the first time, this authors designed triple transgenic mouse: Kera rtTA ; Tet-O-Cre; Smad4 flox/flox (Smad4 kera-cko ),and provided evidence that Smad4 kera-cko mouse developed CE like pathogenesis. The results also indicated that the interface between corneal epithelium and central stoma are smooth in the Smad4 kera-cko group, while those in  the Tgfbr2kera-cko  are irregular. The Smad4 kera-cko mouse is innovative model because corneal Hydrops Cannot be induced by eye-rubbing in Smad4 kera-cko mouse, different with those of Tgfbr2kera-cko mouse.

3.     Fig.6  what specific marker are used in this figure

Author Response

The purpose of this study is to elucidate whether SMAD4-dependent signaling pathway is involved in the TGFBR2-related CE-like pathogenesis. The results indicate that SMAD4-dependent signaling pathway involves in the pathogenesis of CE-like phenotype . 

These new findings provided new knowledge for the mechanism of CE disease like keratoconus, which is a blind causing disease.

The manuscript is well written, the data could support the conclusion.

1.     The main question addressed by the research is that SMAD4-dependent signaling pathway is involved in the TGFBR2-related CE-like pathogenesis.

We appreciate reviewer’s comment and the insight.

2.     For the first time, this authors designed triple transgenic mouse: Kera rtTA ; Tet-O-Cre; Smad4 flox/flox (Smad4kera-cko ),and provided evidence that Smad4kera-cko mouse developed CE like pathogenesis. The results also indicated that the interface between corneal epithelium and central stoma are smooth in the Smad4 kera-cko group, while those in the Tgfbr2kera-cko are irregular. The Smad4kera-cko mouse is innovative model because corneal Hydrops Cannot be induced by eye-rubbing in Smad4kera-cko mouse, different with those of Tgfbr2kera-cko mouse.

We appreciate reviewer’s comment and the insight.

3.     Fig.6 what specific marker are used in this figure?

We appreciate reviewer’s notice, and we are feeling sorry for the confusion, the specific marker is k12.

Reviewer 3 Report

Comments and Suggestions for Authors

Major comments:

1.    The objective of this study is to investigate the involvement of the SMAD4-dependent signaling pathway in TGFBR2-related CE-like pathogenesis by utilizing a novel transgenic mouse strain, Smad4kera-cko, wherein conditional knockout of Smad4 occurs specifically in keratocytes positive for keratocan (Kera). However, the primary focus of the study lies on examining the corneal phenotypes and histological changes associated with the SMAD4-dependent signaling pathway. The molecular mechanisms underlying this pathway or specific interactions between SMAD4 and TGFBR2 are not explored, and no investigation is conducted regarding the functional consequences of the CE-like phenotype observed in the Smad4kera-cko mice.

2.    The authors posit that the similarities observed between Tgfbr2kera-cko and Smad4kera-cko imply the involvement of SMAD4 signaling in the development of corneal thinning phenotype seen in Tgfbr2kera-cko mice. It is important to note, however, that the exact molecular mechanism underlying how increased stromal cell density leads to corneal thinning in Smad4Kera-CKO mice was not investigated in the presented data. Further research is necessary to elucidate these mechanisms before drawing such a conclusion.

Author Response

  1. The objective of this study is to investigate the involvement of the SMAD4-dependent signaling pathway in TGFBR2-related CE-like pathogenesis by utilizing a novel transgenic mouse strain, Smad4kera-cko, wherein conditional knockout of Smad4 occurs specifically in keratocytes positive for keratocan (Kera). However, the primary focus of the study lies on examining the corneal phenotypes and histological changes associated with the SMAD4-dependent signaling pathway. The molecular mechanisms underlying this pathway or specific interactions between SMAD4 and TGFBR2 are not explored, and no investigation is conducted regarding the functional consequences of the CE-like phenotype observed in the Smad4kera-cko mice.

We thank reviewer’s valuable comment and the insight of consideration. SMAD4-dependent signaling pathway is one of the major pathways under the control of TGFBR2 and the major function of SMAD4-dependent signaling is the collagen type-1 formation in corneal stroma. Our previous study using Tgfbr2kera-cko was revealed with less abundant of collagen type 1 in the corneal stroma which served as one of the mechanisms of the corneal stroma thinning. However, in the previous article, we do not know whether such phenotype is due to the decrease of the collagen formation or an enhance of collagen degradation. In this study, to solidate the involvement of SMAD4-dependent signaling and answer reviewer’s comment, we add the experiments of collagen type 1 immunostaining in Smad4kera-cko and Trichrome stain in both Smad4kera-cko and Tgfbr2kera-cko. The result indicated that the stroma thinning was resulted from the collagen formation in both mouse lines and the data can fit each other. On the other hand, for the functional consequences, we performed the OCE to check the corneal stiffness and humbly add the following statement. “The TEM and OCE revealed a decrease of the collagen density and a decrease of the corneal stiffness respectively. Considering that the iCare Tonolab did not show a decrease reading of IOP in Smad4kera-cko, the real IOP under a softer cornea might be higher. An increased IOP and the decreased collagen density which cause the softer cornea in the Smad4kera-cko might serve as the mechanism of a CE-like phenotype at postnatal corneal development.”.   

2. The authors posit that the similarities observed between Tgfbr2kera-cko and Smad4kera-cko imply the involvement of SMAD4 signaling in the development of corneal thinning phenotype seen in Tgfbr2kera-cko mice. It is important to note, however, that the exact molecular mechanism underlying how increased stromal cell density leads to corneal thinning in Smad4Kera-CKO mice was not investigated in the presented data. Further research is necessary to elucidate these mechanisms before drawing such a conclusion.

      We appreciate reviewer’s comment and the insight. We humbly indicated that the increased stromal cell density is the consequence rather than the cause of the stroma thinning. The stroma thinning in Smad4kera-cko is due to the decreased collagen formation in the stroma. However, unlike the Tgfbr2kera-cko mice, knockout the Smad4 in keratocyte will not affect the keratocyte surviving which resulted in the higher density of keratocyte in corneal stroma. Though the density of keratocyte is increased, the number of keratocyte in Smad4kera-cko still lower than control in each section.

Reviewer 4 Report

Comments and Suggestions for Authors

The manuscript entitled "SMAD4-dependent Signaling Pathway Involves in the Pathogenesis of TGFBR2-related CE-like Phenotype" is an interesting study carried out by the authors; however, the authors need to refine the manuscript to be acceptable for publication.

1) The authors need to improve all figures, especially the plots. 

2) The TEM methodology part is missing in the methods section

3) How many samples were analyzed for TEM and how the collagen diameters were measured, the authors need to explain this in the manuscript.

4) Also did the authors looked into the collagen fiber organization pattern using SEM ? It would be interesting to note the organization as well.

5) Also the authors should plot the distribution of collagen fibers between those two experimental condition to see the trend.

Comments on the Quality of English Language

Just require minor edits 

Author Response

The manuscript entitled "SMAD4-dependent Signaling Pathway Involves in the Pathogenesis of TGFBR2-related CE-like Phenotype" is an interesting study carried out by the authors; however, the authors need to refine the manuscript to be acceptable for publication.

1) The authors need to improve all figures, especially the plots. 

We appreciate reviewer’s comment, and we revised the figures with plots.

2) The TEM methodology part is missing in the methods section

We appreciate reviewer’s notice, and the methodology is revised.

3) How many samples were analyzed for TEM and how the collagen diameters were measured, the authors need to explain this in the manuscript.

We appreciate reviewer’s insight. 6 samples were analyzed, and the diameter measurement were measured by the TEM built in system measurement. Manuscript is revised accordingly.

4) Also did the authors looked into the collagen fiber organization pattern using SEM ? It would be interesting to note the organization as well.

We appreciate reviewer’s consideration. In this article, we did not perform the SEM in the collagen fiber organization at this moment since the diameter remains the same. However, in our future article using Tgfbr1, we will take the suggestion and perform the SEM to see if we can have some novel findings.

5) Also, the authors should plot the distribution of collagen fibers between those two experimental condition to see the trend.

We appreciate reviewer’s consideration. We humbly think reviewer was asking the plot of the collagen fiber diameter plot in statistics, so we add the plot in collagen density and diameter calculation.

Reviewer 5 Report

Comments and Suggestions for Authors

Review on Cells-2810792

TGFb signaling is an important pathway leading to homeostasis of mesenchymal cells and related tissues like corneal stroma, thus defects of TGFb were hypothesized to be one of the causes of corneal ectasia (CE).  The authors successfully recapitulated this CE phenotype in mice where the Tgfbr2, the major component of TGFb pathway, is conditionally deleted in keratocytes in the corneal stroma in their previous report. In current manuscript, the authors continue to search for downstream components in the pathway linked to the CE phenotype by knocking out the Smad4 gene in keratocytes in the cornea. It is an interesting work and story is well written. However, questions whether Smad4 is truly deleted in the keratocytes and whether some of their measurements are appropriate are remained unanswered. All these questions need be clarified before the manuscript is qualified for publication.

Major concerns:

1.       Please clarify whether TGFb is one of the genes identified for cornea ectasia in clinics?

2.       All mouse strains and Dox administration needed for making Smad4 kera-cKO need be detailed in the Methods.

3.       The confirmation of Smad4 deletion in the keratocytes in the cornea must be presented before any study started using such cKO mice. My suggestion for this is to isolate the corneas from both control and Kera-cKO mice where the epithelium is mechanically removed and saved for positive control for expression of Smad4. The isolated corneal tissues without the epithelium are utilized either for genomic DNA extraction for genotyping for Smad4 exon 8 as the authors indicated, or for total protein extraction for Western blot analysis for Smad4 protein presence. For either approach, the corneal epithelial tissues (or other tissue like tail tips) are utilized as positive control for Smad4 expression.

4.       In the results, all comparisons between the control and Kera-cKO must be under the same condition. For example, the pictures in figure 2C (Ctrl) and 2D (Kera-cKO) are seemingly not at the same amplification scale. The other example is the images in figure 3A through 3F where the control pupils were obviously dilated whereas the Kera-cKO eyes were not dilated.

Minor suggestions:

1.       Any abbreviations must be spelt out when first appear in the text, may need repeat if needed such as CE, IOP, and TEM in the abstract.

2.       The primer sets for Kera-rtTA and TC need be listed in detail in the Genotyping section.

3.       For statistic analyses, ANOVA should be used when Ctrl vs. cKO and Tgfb2 cKO vs. Smad4 cKO are compared simultaneously in figure 1.

4.       The sample size in all bar graphs must be indicated. If n=1 like figure 2F, the result is not reliable thereby should be removed.

5.       In Result 3.3 “Visual Acuity (VA)” is mistakenly used because there was no direct detection of visual acuity here.

6.       What Fig. E-L is referred to?

7.       Please make sure there is such a so-called basal membrane in mouse cornea.

8.       I do not like figures 4 to 6 where no quantification is presented.

Author Response

TGFb signaling is an important pathway leading to homeostasis of mesenchymal cells and related tissues like corneal stroma, thus defects of TGFb were hypothesized to be one of the causes of corneal ectasia (CE).  The authors successfully recapitulated this CE phenotype in mice where the Tgfbr2, the major component of TGFb pathway, is conditionally deleted in keratocytes in the corneal stroma in their previous report. In current manuscript, the authors continue to search for downstream components in the pathway linked to the CE phenotype by knocking out the Smad4 gene in keratocytes in the cornea. It is an interesting work and story is well written. However, questions whether Smad4 is truly deleted in the keratocytes and whether some of their measurements are appropriate are remained unanswered. All these questions need be clarified before the manuscript is qualified for publication.

We appreciate the reviewer’s question, idea, insight, and consideration very much. To address the major question above, we perform the SMAD4 immunostaining and confirmed that the Smad4 was successfully deleted in the keratocytes. 

Major concerns:

  1. Please clarify whether TGFb is one of the genes identified for cornea ectasia in clinics?

We appreciate the reviewer’s insight. We humbly add the following statement “Our previous findings are consistence with genomic and RNA-seq studies in clinics, which implicated that TGFB signaling dysfunction might be associated with the etiology of CE. (6)” in the manuscript.

  1. All mouse strains and Dox administration needed for making Smad4kera-cKO need be detailed in the Methods.

We appreciate the reviewer’s notice. The methodology is revised accordingly.

  1. The confirmation of Smad4 deletion in the keratocytes in the cornea must be presented before any study started using such cKO mice. My suggestion for this is to isolate the corneas from both control and Kera-cKO mice where the epithelium is mechanically removed and saved for positive control for expression of Smad4. The isolated corneal tissues without the epithelium are utilized either for genomic DNA extraction for genotyping for Smad4 exon 8 as the authors indicated, or for total protein extraction for Western blot analysis for Smad4 protein presence. For either approach, the corneal epithelial tissues (or other tissue like tail tips) are utilized as positive control for Smad4 expression.

We appreciate the reviewer’s consideration and suggestion very much. To address this question above, we humbly and carefully perform an alternative experiment which is the SMAD4 immunostaining in the whole cornea, and the data indicated that the SMAD4 protein can be detected in the corneal epithelium from both experimental group and control group and can be detected in the Smad4Ctrl corneal stroma but CANNOT be detected in the corneal stroma in Smad4kera-cko. The SMAD4 immunostaining humbly confirmed that the Smad4 was successfully deleted in the keratocytes of Smad4kera-cko mice. 

  1. In the results, all comparisons between the control and Kera-cKO must be under the same condition. For example, the pictures in figure 2C (Ctrl) and 2D (Kera-cKO) are seemingly not at the same amplification scale. The other example is the images in figure 3A through 3F where the control pupils were obviously dilated whereas the Kera-cKO eyes were not dilated.

We appreciate the reviewer’s consideration and notice. For the Fig. 2C and 2D, we believe that they should be in the same scale and the tremendous protrusion feeling might be due to the enlarged eyelid open size resulted from the corneal ectasia. However, at this moment, due to the original data and the machine setting of the mouse positioning are stored in another university and the scale cannot be found, we decided to take this data away for maintaining the quality. For the pupil’s dilation concern, usually with enough anesthesia time, the pupil will dilate wider. However, considering that the surface condition in the Smad4kera-cko is not as good as Smad4Ctrl and the tear film is easy to break up in Smad4kera-cko, which make the light sources measurement harder, we performed the picturing once the Smad4kera-cko mice fall asleep. On the other hand, the pupil color and structure can provide us a good background for checking the corneal transparency at the same time, so we keep this data. In fear and trepidation, considering that the pupil size is not our aiming data, we humbly hope reviewer can accept this set of data since the original machine setting for the measurement is in another university. 

Minor suggestions:

  1. Any abbreviations must be spelt out when first appear in the text, may need repeat if needed such as CE, IOP, and TEM in the abstract.

We appreciate the reviewer’s consideration, and we revised the manuscript accordingly.

  1. The primer sets for Kera-rtTA and TC need be listed in detail in the Genotyping section.

We appreciate the reviewer’s comment, and we revised the methodology accordingly.

  1. For statistic analyses, ANOVA should be used when Ctrl vs. cKO and Tgfb2 cKO vs. Smad4 cKO are compared simultaneously in figure 1.

      We appreciate the reviewer’s consideration and comments. The litters of these two set statistics are not from the same litter and this is why we performed the t-test separately for a simple comparison.

  1. The sample size in all bar graphs must be indicated. If n=1 like figure 2F, the result is not reliable thereby should be removed.

We appreciate the reviewer’s consideration. The n = 6 in the Tonolab IOP measurement. 

  1. In Result 3.3 “Visual Acuity (VA)” is mistakenly used because there was no direct detection of visual acuity here.

We appreciate the reviewer’s consideration, and we understand that VA is a term used in the clinics and the diagnosis is based on the vision test. To avoid the confusion, we only use the term of VA when describe the human patients and use another term called VP (vision precision) when describing mouse situation.

  1. What Fig. E-L is referred to?

We appreciate the reviewer’s notice and check all the Fig. rationale accordingly.

  1. Please make sure there is such a so-called basal membrane in mouse cornea.

We appreciate the reviewer’s notice and check the text accordingly.

  1. I do not like figures 4 to 6 where no quantification is presented.

We appreciate the reviewer’s comment and consideration. We performed the statistics accordingly to those data can be quantified.

Round 2

Reviewer 1 Report

Comments and Suggestions for Authors

In this revised manuscript, the authors have added some experimental data from Tgfbr2kera-cko mice. The results showed similar changes in collagen type 1 immunostaining in the cornea between Tgfbr2kera-cko and Smad4kera-cko mice. I agree with the authors that SMAD4-dependent signaling involves corneal ectasia. However, the similar changes between Tgfbr2kera-cko and Smad4kera-cko mice are not sufficient to verify the relationship between Tgfbr2 and Smad4 in corneal ectasia. The study was flawed in its design.

Comments on the Quality of English Language

Please revise English grammar.

Reviewer 3 Report

Comments and Suggestions for Authors

The authors have appropriately addressed most of the comments and this revised paper has been improved over its original version. I have no more major concerns and feel it is now acceptable for publication.

Reviewer 4 Report

Comments and Suggestions for Authors

Thanks for addressing the comments. 

Reviewer 5 Report

Comments and Suggestions for Authors

All my major concerns are fully addressed although the presence of Smad4 in the control corneal stroma is hardly seen in Figure 4A. Anyway, it should be qualified for publication in Cells now.